# AN AGNOSTIC VIEW ON THE COST OF OVERFITTING IN (KERNEL) RIDGE REGRESSION

**Lijia Zhou**
University of Chicago
zlj@uchicago.edu

**James B. Simon**
UC Berkeley and Generally Intelligent
james.simon@berkeley.edu

**Gal Vardi**
TTI-Chicago and Hebrew University
galvardi@ttic.edu

**Nathan Srebro**
TTI-Chicago
nati@ttic.edu

## ABSTRACT

We study the cost of overfitting in noisy kernel ridge regression (KRR), which we define as the ratio between the test error of the interpolating ridgeless model and the test error of the optimally-tuned model. We take an "agnostic" view in the following sense: we consider the cost as a function of sample size for any target function, even if the sample size is not large enough for consistency or the target is outside the RKHS. We analyze the cost of overfitting under a Gaussian universality ansatz using recently derived (non-rigorous) risk estimates in terms of the task eigenstructure. Our analysis provides a more refined characterization of benign, tempered and catastrophic overfitting (cf. Mallinar et al., 2022).

## 1 INTRODUCTION

The ability of large neural networks to generalize, even when they overfit to noisy training data (Neyshabur et al., 2015; Zhang et al., 2017; Belkin et al., 2019), has significantly challenged our understanding of the effect of overfitting. A starting point for understanding overfitting in deep learning is to understand the issue in kernel methods, possibly viewing deep learning through their kernel approximation (Jacot et al., 2020). Indeed, there is much progress in understanding the effect of overfitting in kernel ridge regression and ridge regression with Gaussian data. It has been shown that the test error of the minimal norm interpolant can approach Bayes optimality and so overfitting is "benign" (Bartlett et al., 2020; Muthukumar et al., 2020; Koehler et al., 2021; Wang et al., 2021; Donhauser et al., 2022). In other situations such as Laplace kernels and ReLU neural tangent kernels, the interpolating solution is not consistent but also not "catastrophically" bad, which falls into an intermediate regime called "tempered" overfitting (Mallinar et al., 2022).

However, the perspective taken in this line of work differs from the agnostic view of statistical learning. These results typically focus on asymptotic behavior and consistency of a well-specified model, asking how the limiting behavior of interpolating learning rules compares to the Bayes error (the smallest risk attainable by any measurable function of the feature $x$). In contrast, the agnostic PAC model (Vapnik & Chervonenkis, 1971; Haussler, 1992; Shalev-Shwartz & Ben-David, 2014) does not require any assumption on the conditional distribution of the label $y$. In particular, the conditional expectation $\mathbb{E}[y|x]$ is not necessarily a member of the hypothesis class and it does not need to have small Hilbert norm in the Reproducing Kernel Hilbert Space (RKHS). Instead, the learning rule is asked to find a model whose test risk can compete with the smallest risk *within* the hypothesis class, which can be quite high if no predictor in the hypothesis class can attain the Bayes error. In these situations, the agnostic PAC model can still provide a meaningful learning guarantee.

Furthermore, we would like to isolate the effect of overfitting (i.e. underregularizing, and choosing to use a predictor that fits the noise, instead of compromising on empirical fit and choosing a predictor that balances empirical fit with complexity or norm) from the difficulty of the learning problem and appropriateness of the model irrespective of overfitting (i.e. even if we were to not overfit and instead optimally balance empirical fit and norm, as in ridge regression). A view which considers

only the risk of the overfitting rule (e.g. Mallinar et al., 2022) confounds these two issues. Instead, we would like to study the direct effect of overfitting: how much does it hurt to overfit and use ridgeless regression *compared to* optimally tuned ridge regression.

In this paper, we take an agnostic view to the direct effect of overfitting in (kernel) ridge regression. Rather than comparing the asymptotic risk of the interpolating ridgeless model to the Bayes error, we compare it to the best ridge model in terms of population error as a function of sample size, and we measure the cost of overfitting as a ratio. We show that the cost of overfitting can be bounded by using only the sample size and the effective ranks of the covariance, even when the risk of the optimally-tuned model is high relative to the Bayes error. Our analysis applies to any target function (including ones with unbounded RKHS norm) and recovers the matching upper and lower bounds from Bartlett et al. (2020), which allows us to have a more refined understanding of the benign overfitting. In addition to benign overfitting, we show that the amount of "tempered" overfitting can also be understood using the cost of interpolation, and we derive the necessary and sufficient condition for "catastrophic" overfitting (Mallinar et al., 2022). Combining these results leads to a refined notion of benign, tempered, and catastrophic overfitting (focusing on the difference versus the optimally tuned predictor), and a characterization as a function of sample size $n$ based on computing the effective rank $r_k$ at some index $k$. We further apply our results to the setting of inner product kernels in the polynomial regime (Ghorbani et al., 2021; Mei et al., 2022; Misiakiewicz, 2022) and recover the multiple descent curve.

## 2 PROBLEM FORMULATION

Let $\mathcal{X}$ be an abstract input space and $K : \mathcal{X} \times \mathcal{X} \to \mathbb{R}$ a positive semi-definite kernel[1].

### 2.1 BI-CRITERION OPTIMIZATION IN KRR

Given a data set $D_n$ consisting of $(x_1, y_1), ..., (x_n, y_n) \in \mathcal{X} \times \mathbb{R}$ sampled from some unknown joint distribution $\mathcal{D}$, in order to find a predictor with good test error $R(f)$, we solve the bi-criterion optimization:

$$\min_{f \in \mathcal{H}} \hat{R}(f), \|f\|_{\mathcal{H}} \tag{1}$$

where $\|f\|_{\mathcal{H}}$ is the Hilbert norm in the RKHS and the test error and training error (in square loss) of a predictor $f$ is given by

$$R(f) := \mathbb{E}\left[(f(x) - y)^2\right] \quad \text{and} \quad \hat{R}(f) := \frac{1}{n} \sum_{i=1}^{n} (f(x_i) - y_i)^2.$$

The Pareto-frontier of the bi-criterion problem (1) corresponds to the regularization path $\{\hat{f}_\delta\}_{\delta \geq 0}$ given by the sequence of problems:

$$\hat{f}_\delta = \arg\min_{f \in \mathcal{H}} \hat{R}(f) + \frac{\delta}{n} \|f\|_{\mathcal{H}}^2.$$

By the representation theorem, $\hat{f}_\delta$ has the explicit closed form:

$$\hat{f}_\delta(x) = K(D_n, x)^T \left(K(D_n, D_n) + \delta I_n\right)^{-1} Y \tag{2}$$

where $K(D_n, x) \in \mathbb{R}^n, K(D_n, D_n) \in \mathbb{R}^{n \times n}, Y \in \mathbb{R}^n$ are given by $[K(D_n, x)]_i = K(x_i, x)$, $[K(D_n, D_n)]_{i,j} = K(x_i, x_j)$ and $[Y]_i = y_i$. The interpolating "ridgeless" solution (minimal norm interpolant) is the extreme Pareto point and obtained by taking $\delta \to 0^+$:

$$\hat{f}_0 = \arg\min_{f \in \mathcal{H}: \hat{R}(f) = 0} \|f\|_{\mathcal{H}}.$$

Even though $\hat{f}_0$ has the minimal norm among all interpolants, the norm of $\hat{f}_0$ will still be very large because it needs to memorize all the noisy training labels. In this paper, we are particularly interested in the generalization performance of the ridgeless solution $\hat{f}_0$, which minimizes the training error in the bi-criterion problem (1) too much.

---

[1]i.e.: (i) $\forall x, x' \in \mathcal{X}, \ K(x, x') = K(x', x)$, and (ii) $\forall n \in \mathbb{N}, \ x_1, ..., x_n \in \mathcal{X}, \ c_1, ..., c_n \in \mathbb{R}$, it holds that $\sum_{i=1}^{n} \sum_{j=1}^{n} c_i c_j K(x_i, x_j) \geq 0$.

## 2.2 Mercer's Decomposition

Though the setting for KRR is very generic, we can understand it as (linear) ridge regression. By Mercer's theorem (Mercer, 1909), the kernel admits the decomposition

$$K(x, x') = \sum_i \lambda_i \phi_i(x) \phi_i(x') \tag{3}$$

where $\phi_i : \mathcal{X} \to \mathbb{R}$ form a complete orthonormal basis satisfying $\mathbb{E}_x[\phi_i(x)\phi_j(x)] = 1$ if $i = j$ and 0 otherwise, and the expectation is taken with respect to the marginal distribution of $x$ given by $\mathcal{D}$. For example, if $\mathcal{X} = \{x_1, ..., x_M\}$ has finite cardinality $M$ and $x$ is uniformly distributed over $\mathcal{X}$, then (3) can be found by the spectral decomposition of the matrix $K(\mathcal{X}, \mathcal{X}) \in \mathbb{R}^{M \times M}$ given by $[K(\mathcal{X}, \mathcal{X})]_{i,j} = K(x_i, x_j)$. When $x$ is uniformly distributed over the sphere in $\mathbb{R}^d$ or the boolean hypercube $\{-1, 1\}^d$, then $\{\phi_i\}$ can be taken to be the spherical harmonics or the Fourier-Walsh (parity) basis. In the case that $K$ is the Gaussian kernel or polynomial kernel, the eigenvalues $\{\lambda_i\}$ has closed-form expression in terms of the modified Bessel function or the Gamma function (Minh et al., 2006).

Therefore, instead of viewing the feature $x$ as an element of $\mathcal{X}$, we can consider the potentially infinite-dimensional real-valued vector $\psi(x) = (\sqrt{\lambda_1}\phi_1(x), \sqrt{\lambda_2}\phi_2(x), ...)$ and denote the design matrix $\Psi = [\psi(x_1), \psi(x_2), ...]^T$. Then we can write $K(x, x') = \langle \psi(x), \psi(x') \rangle$ and understand the predictor in (2) as

$$\hat{f}_\delta(x) = \psi(x)^T \Psi^T (\Psi \Psi^T + \delta I_n)^{-1} Y$$
$$= \langle \hat{w}_\delta, \psi(x) \rangle$$

where $\hat{w}_\delta = \Psi^T (\Psi \Psi^T + \delta I_n)^{-1} Y$ is simply the ridge regression estimate with respect to the data set $(\Psi, Y)$. For a predictor $f$ of the form $f(x) = \langle w, \psi(x) \rangle$, its Hilbert norm is given by $\|f\|_{\mathcal{H}} = \|w\|_2$.

The Bayes-optimal target function is $f_*(x) = \mathbb{E}_{(x,y)\sim\mathcal{D}}[y|x]$. We may expand this function in the kernel eigenbasis as $f_*(x) = \sum_i v_i \phi_i(x)$, where $\{v_i\}$ are eigencoefficients. Let the *noise level* be $\sigma^2 = \mathbb{E}_{(x,y)\sim\mathcal{D}}[(y - f_*(x))^2]$.

## 2.3 Closed-form Risk Estimate for (Kernel) Ridge Regression

A great number of recent theoretical works have converged on a powerful set of closed-form equations which estimate the test risk of KRR in terms of task eigenstructure (Hastie et al., 2019; Wu & Xu, 2020; Jacot et al., 2020; Canatar et al., 2021; Loureiro et al., 2021; Mel & Ganguli, 2021; Richards et al., 2021). We shall use the risk estimate from these works as our starting point. These equations rely on (some variant of) the following Gaussian design ansatz:

**Assumption 1** (Gaussian design ansatz). When sampling $(x, \cdot) \sim \mathcal{D}$, the eigenfunctions are either *Gaussian* in the sense that $\psi(x) \sim \mathcal{N}(0, \text{diag}(\{\lambda_i\}))$, or else we have *Gaussian universality* in the sense that the expected test risk is unchanged if we replace $\psi(x)$ with $\tilde{\psi}(x)$, where $\tilde{\psi}$ is Gaussian in this manner.

Remarkably, Assumption 1 appears to hold even for many *real datasets*: predictions computed for Gaussian design agree excellently with kernel regression experiments with real data (Canatar et al., 2021; Simon et al., 2021; Wei et al., 2022). We will take Assumption 1 henceforth.

We now state the "omniscient risk estimate" presented by this collection of works.[2] First, let the *effective regularization constant* $\kappa_\delta$ be the unique nonnegative solution to

$$\sum_i \frac{\lambda_i}{\lambda_i + \kappa_\delta} + \frac{\delta}{\kappa_\delta} = n. \tag{4}$$

Using $\kappa_\delta$, we can define

$$\mathcal{L}_{i,\delta} = \frac{\lambda_i}{\lambda_i + \kappa_\delta} \quad \text{and} \quad \mathcal{E}_\delta = \frac{n}{n - \sum_i \mathcal{L}_{i,\delta}^2}, \tag{5}$$

---

[2]We adopt the notation of Simon et al. (2021), but the risk estimates of all mentioned works are equivalent. We take the term "omniscient risk estimate" from Wei et al. (2022).

where we refer to $\mathcal{L}_{i,\delta}$ as the *learnability of mode* $i$ and $\mathcal{E}_\delta$ as the *overfitting coefficient*. The expected test risk over datasets is then given approximately by

$$R(\hat{f}_\delta) \approx \tilde{R}(\hat{f}_\delta) := \mathcal{E}_\delta \left( \sum_i (1 - \mathcal{L}_{i,\delta})^2 v_i^2 + \sigma^2 \right). \tag{6}$$

The "$\approx$" in (6) can be given several meanings. Firstly, it becomes an equivalence in an appropriate asymptotic limit in which $n$ and the number of eigenmodes in a given eigenvalue range both grow proportionally large (Hastie et al., 2019; Bach, 2023). Secondly, with fixed task eigenstructure, the error incurred can be bounded by a decaying function of $n$ (Cheng & Montanari, 2022). Thirdly, numerical experiments attest that the error is small even at quite modest $n$ (Canatar et al., 2021; Simon et al., 2021). For the rest of this paper, we will simply treat it as an equivalence, formally proving facts about the omniscient risk estimate $\tilde{R}(\hat{f}_\delta)$. Thus, our results follow by analyzing the expression from (6).

## 3 COST OF OVERFITTING

The sensible and traditional approach to learning using a complexity penalty, such as the Hilbert norm $\|f\|_{\mathcal{H}}$, is to use a Pareto point (point on the regularization path) of the bi-criteria problem (1) that minimizes some balanced combination of the empirical risk and penalty (the "structural risk") so as to ensure small population risk. Assumptions about the problem can help us choose which Pareto optimal point, i.e. what value of the tradeoff parameter $\delta$, to use. Simpler and safer is to choose this through validation: calculate the Pareto frontier (aka regularization path) on half the training data set, and choose among these Pareto points by minimizing the "validation error" on the held-out half of the training set. Here we do not get into these details, and simply compare to the best Pareto point:

$$R(\hat{f}_{\delta^*}) = \inf_{\delta \geq 0} R(\hat{f}_\delta).$$

Although we cannot find $\hat{f}_{\delta^*}$ exactly empirically, it is useful as an oracle, and studying the gap versus this ideal Pareto point provides an upper bound on the gap versus any possible Pareto point (i.e. with any amount of "ideal" regularization). And in practice, as well as theoretically, a validation approach as described above will behave very similar to $\hat{f}_{\delta^*}$. We therefore define the **cost of overfitting** as the (multiplicative) gap between the interpolating predictor $\hat{f}_0$ and the optimaly regularized $\hat{f}_{\delta^*}$:

**Definition 1.** Given any data distribution $\mathcal{D}$ over $\mathcal{X} \times \mathbb{R}$ and sample size $n \in \mathbb{N}$, we define the cost of overfitting as

$$C(\mathcal{D}, n) := \frac{R(\hat{f}_0)}{\inf_{\delta \geq 0} R(\hat{f}_\delta)}, \quad \text{and its prediction based on (6): } \quad \tilde{C}(\mathcal{D}, n) := \frac{\tilde{R}(\hat{f}_0)}{\inf_{\delta \geq 0} \tilde{R}(\hat{f}_\delta)}$$

It is possible to directly analyze $R(\hat{f}_0)$ and $R(\hat{f}_{\delta^*})$ (or their predictions based on (6)) in order to study the cost of overfitting. However, any bound on $R(\hat{f}_0)$ or $R(\hat{f}_{\delta^*})$ will necessarily depend on the target function. Instead, we show that there is a much simpler argument to bound the cost of overfitting.

**Theorem 1.** *Consider $\mathcal{E}_0$ defined in (5) with $\delta = 0$, then it holds that*

$$\tilde{C}(\mathcal{D}, n) \leq \mathcal{E}_0. \tag{7}$$

*Proof.* Observe that

$$\tilde{R}(\hat{f}_{\delta^*}) = \inf_{\delta \geq 0} \mathcal{E}_\delta \left( \sum_i (1 - \mathcal{L}_{i,\delta})^2 v_i^2 + \sigma^2 \right)$$

$$\geq \inf_{\delta \geq 0} \sum_i (1 - \mathcal{L}_{i,\delta})^2 v_i^2 + \sigma^2$$

$$= \sum_i (1 - \mathcal{L}_{i,0})^2 v_i^2 + \sigma^2$$

where we use the fact that $(1 - \mathcal{L}_{i,\delta})^2$ decreases as $\kappa_\delta$ decreases, and $\kappa_\delta$ decreases as $\delta$ decreases. The proof concludes by observing $\sum_i (1 - \mathcal{L}_{i,0})^2 v_i^2 + \sigma^2 = \tilde{R}(\hat{f}_0)/\mathcal{E}_0$. $\qquad \square$

Indeed, (4) and (5) used to define $\mathcal{E}_0$ does not depend on the target coefficients. It is also straightforward to check that if $v_i = 0$, then $\tilde{R}(\hat{f}_0) = \mathcal{E}_0 \sigma^2$ and $\tilde{R}(\hat{f}_{\delta^*}) = \sigma^2$ by choosing $\delta^* = \infty$, and $\tilde{C}(\mathcal{D}, n) = \mathcal{E}_0$ for any $n$. This shows that (7) is the tightest agnostic bound on the cost of overfitting:

$$\forall_{P(x)} \; \mathcal{E}_0 = \sup_{P(y|x)} \tilde{C}(\mathcal{D}, n)$$

where $\mathcal{E}_0$ on the left-hand-side depends only on the marginal $P(x)$, while $\tilde{C}(\mathcal{D}, n)$ depends on both the marginal $P(x)$ and the conditional $P(y|x)$.

More generally, it is clear that we have the lower bound

$$\tilde{C}(\mathcal{D}, n) \geq \mathcal{E}_0 \frac{\sigma^2}{\tilde{R}(\hat{f}_{\delta^*})}$$

due to the non-negativity of $v_i^2$ in (6). Thus, from the above and Theorem 1, we have $\frac{\sigma^2}{\tilde{R}(\hat{f}_{\delta^*})} \leq \frac{\tilde{C}(\mathcal{D},n)}{\mathcal{E}_0} \leq 1$. Therefore, if $\frac{\sigma^2}{\tilde{R}(\hat{f}_{\delta^*})} \to 1$ as $n \to \infty$, namely, the optimal-tuned ridge is consistent, then $\frac{\tilde{C}(\mathcal{D},n)}{\mathcal{E}_0} \to 1$. That is, in this case $\mathcal{E}_0$ precisely captures the cost of overfitting.

If the optimal-tuned ridge is not consistent, (7) might be a loose upper bound on $\tilde{C}(\mathcal{D}, n)$. However, under our assumption, even in this case $\mathcal{E}_0$ still captures the qualitative noisy overfitting behavior in the following sense: If $\lim_{n\to\infty} \mathcal{E}_0 = 1$, we have benign overfitting, i.e. $\tilde{C} \to 1$, regardless of the target; If $\lim_{n\to\infty} \mathcal{E}_0 = \infty$ and $\sigma^2 > 0$, then we have catastrophic overfitting, i.e. $\tilde{C} \to \infty$, regardless of the target; If $1 < \lim_{n\to\infty} \mathcal{E}_0 < \infty$ then overfitting is either benign or tempered.

Finally, we note that the argument in the proof of Theorem 1 shows something more: for any $\delta \leq \delta^*$, it holds that $\tilde{R}(\hat{f}_\delta) \leq \mathcal{E}_\delta \tilde{R}(\hat{f}_{\delta^*}) \leq \mathcal{E}_0 \tilde{R}(\hat{f}_{\delta^*})$. Therefore, the quantity $\mathcal{E}_0$ bounds the cost of overfitting not only for the interpolating solution, but also for any ridge model with a sufficiently small regularization parameter $\delta$. Consequently, if $\mathcal{E}_0$ is close to one, then the risk curve will become flat once all of the signal is fitted (for example, see Figure 1 of Zhou et al. (2021)), exhibiting the double descent phenomenon instead of the classical U-shape curve (Belkin et al., 2019). Similar results on the flatness of the generalization curve are proven in Tsigler & Bartlett (2020) and Zhou et al. (2021).

### 3.1 BENIGN OVERFITTING

In this section, we discuss when $\mathcal{E}_0$ can be close to 1 and so overfitting is benign. Note that the target coefficients play no role at all in our analysis. To further upper bound the cost of overfitting, we will introduce the notion of effective rank (Bartlett et al., 2020).

**Definition 2.** The effective ranks of a covariance matrix with eigenvalues $\{\lambda_i\}_{i=1}^\infty$ in descending order are defined as

$$r_k = \frac{\sum_{i>k} \lambda_i}{\lambda_{k+1}} \qquad \text{and} \qquad R_k := \frac{\left(\sum_{i>k} \lambda_i\right)^2}{\sum_{i>k} \lambda_i^2}.$$

The two effective ranks are closely related to each other by the relationship $r_k \leq R_k \leq r_k^2$ and are equal if $\Sigma$ is the identity matrix (Bartlett et al., 2020). Roughly speaking, the minimal norm interpolant can approximate the target in the span of top $k$ eigenfunctions and use the remaining components of $x$ to memorize the residual. A large effective rank ensures that the small eigenvalues of $\Sigma$ are roughly equal to each other and so it is possible to evenly spread out the cost of overfitting into many different directions. More precisely, we show the following finite-sample bound on $\mathcal{E}_0$, which decreases to 1 as $n$ increases if $k = o(n)$ and $R_k = \omega(n)$:

**Theorem 2.** *For any $k < n$, it holds that*

$$\mathcal{E}_0 \leq \left(1 - \frac{k}{n}\right)^{-2} \left(1 - \frac{n}{R_k}\right)_+^{-1}. \tag{8}$$

The conditions that $k = o(n)$ and $R_k = \omega(n)$ are two key conditions for benign overfitting in linear regression (Bartlett et al., 2020). They require an additional assumption that $r_0 = o(n)$ for

consistency, which is sufficient for the consistency of the optimally tuned model when the target is well-specified. Our Theorem 2 provides a more refined understanding of benign overfitting: at a finite sample $n$, if we can choose a small $k$ such that $R_k$ is large relative to $n$, then the interpolating ridgeless solution is nearly as good as the optimally tuned model, regardless of whether the optimally tuned model can learn the target. Furthermore, we also recover a version of the matching lower bound of Theorem 4 in Bartlett et al. (2020), though our proof technique is completely different and simpler since we have a closed-form expression. Since $\mathcal{E}_0 = \left(1 - \frac{1}{n}\sum_i \mathcal{L}_{i,0}\right)^{-1}$, it suffices to lower bound $\frac{1}{n}\sum_i \mathcal{L}_{i,0}^2$.

**Theorem 3.** *Fix any $b > 0$. If there exists $k < n$ such that $n \le k + br_k$, then let $k$ be the first such integer. Otherwise, pick $k = n$. It holds that*

$$\frac{1}{n}\sum_i \mathcal{L}_{i,0}^2 \ge \max\left\{\frac{1}{(b+1)^2}\left(1 - \frac{k}{n}\right)^2 \frac{n}{R_k}, \left(\frac{b}{b+1}\right)^2 \frac{k}{n}\right\}. \tag{9}$$

For simplicity, we can take $b = 1$ in the lower bound above. We see that $\mathcal{E}_0$ cannot be close to 1 unless $k$ is small relative to $n$. Even if $k$ is small, the first term in (9) requires $n/R_k$ to be small. Conversely, if both $k/n$ and $n/R_k$ are small, then we can apply Theorem 2 to show that $\mathcal{E}_0$ is close to 1 and we have identify the necessary and sufficient condition for $\mathcal{E}_0 \to 1$.

**Corollary 1.** *For any $n \in \mathbb{N}$, let $k_n$ be the first integer $k < n$ such that $n \le k + r_k$. Then $\mathcal{E}_0 \to 1$ if and only if*

$$\lim_{n\to\infty} \frac{k_n}{n} = 0 \quad and \quad \lim_{n\to\infty} \frac{n}{R_{k_n}} = 0. \tag{10}$$

Though Corollary 1 is stated as an asymptotic result, the spectrum is allowed to change with the sample size $n$ and the target function plays no role in condition (10). Next, we apply our results to some canonical examples where overfitting is benign.

**Example 1** (Benign covariance from Bartlett et al. (2020)).

$$\lambda_i = i^{-1}\log^{-\alpha} i \quad \text{for some } \alpha > 0.$$

In this case, we can estimate

$$\sum_{i>k}\lambda_i \ge \int_{k+1}^\infty \frac{1}{x\log^\alpha x}\,dx = \frac{1}{(\alpha - 1)\log^{\alpha-1}(k+1)}$$

$$\sum_{i>k}\lambda_i^2 \le \frac{1}{k+1}\int_k^\infty \frac{1}{x\log^{2\alpha} x}\,dx = \frac{1}{(k+1)(2\alpha - 1)\log^{2\alpha-1}(k)}$$

and so

$$R_k \ge \frac{(k+1)(2\alpha - 1)\log^{2\alpha-1}(k)}{(\alpha - 1)^2\log^{2\alpha-2}(k+1)} = \Theta\left(k\log k\right).$$

Then by choosing $k = \Theta\left(\frac{n}{\sqrt{\log n}}\right)$, we have $k = o(n)$ and $R_k = \omega(n)$ because $\frac{R_k}{n} = \Theta(\log^{1/2} n)$.

**Example 2** (Junk features from Zhou et al. (2020)).

$$\lambda_i = \begin{cases} 1 & \text{if } i \le d_S \\ \frac{1}{d_J} & \text{if } d_S + 1 \le i \le d_S + d_J \\ 0 & \text{if } i > d_S + d_J. \end{cases}$$

In this case, it is routine to check $R_k = d_J$ by choosing $k = d_S$. Letting $d_S = o(n)$ and $d_J = \omega(n)$, Theorem 2 shows that $\mathcal{E}_0 \to 1$.

Finally, we show our bound (8) also applies to isotropic features in the proportional regime even though overfitting is not necessarily benign.

**Example 3** (Isotropic features in the proportional regime).

$$\lambda_i = \begin{cases} 1 & \text{if } i \le d \\ 0 & \text{otherwise} \end{cases} \quad \text{for} \quad d = \gamma n \quad \text{and} \quad \gamma > 1.$$

In this case, it is easy to check that $r_k = d - k$ and so $k + r_k = d > n$ and $k_n = 0$. The first condition in (10) holds because $k_n/n = 0$. However, the second condition in (10) does not hold because $R_k = d - k$ and $n/R_{k_n} = 1/\gamma > 0$. Plugging in $k = 0$ to Theorem 2, we obtain

$$\mathcal{E}_0 \le \left(1 - \frac{n}{d}\right)^{-1} = \frac{\gamma}{\gamma - 1}.$$

The above upper bound is tight when $v_i = 0$ because it is well-known that in the proportional regime (for example, see Hastie et al. (2019) and Zhou et al. (2021)), it holds that

$$\lim_{n\to\infty} R(\hat{f}_0) = \sigma^2 \frac{\gamma}{\gamma - 1}.$$

### 3.2 TEMPERED OVERFITTING

Theorem 2 allows us to understand the cost of overfitting when it is benign. However, it is not informative when no $k < n$ satisfies $R_k > n$. In Theorem 4 below, we provide an estimate for the amount of "tempered" overfitting based on the ratio $k/r_k$ over a finite range of indices.

**Theorem 4.** *Fix any $\epsilon \in (0, n/r_0)$ and consider $k_l, k_u \in \mathbb{N}$ given by*

$$k_l := \max\{k \ge 0 : k + \epsilon r_k \le n\}$$
$$k_u := \min\{k \ge 0 : k + r_k \ge (1 + \epsilon^{-1})n\}.$$

*Then it holds that*

$$\mathcal{E}_0 \le (1 + \epsilon)^2 \cdot \max_{k_l \le k < k_u} \left(\frac{\lambda_{k+1}}{\lambda_{k+2}} + \frac{1}{\epsilon} \frac{k+1}{r_k - 1}\right). \tag{11}$$

To interpret (11), we first suppose that the spectrum $\{\lambda_i\}$ does not change with $n$ and has infinitely many non-zero eigenvalues (which is the case in Example 1, 4 and 5 below). For any fixed $\epsilon > 0$, $k_l$ must increases as $n$ increases. If $k$ is large, then it is usually the case that $\lambda_{k+1} \approx \lambda_k$ or the ratio is bounded. Letting $\epsilon = 1$, we can understand (11) as $\mathcal{E}_0 \lesssim 1 + \frac{k}{r_k}$.

In particular, if $r_k = \Omega(k)$, then $\mathcal{E}_0$ is bounded and overfitting cannot be catastrophic. Conversely, we show that overfitting is catastrophic when $r_k = o(k)$ in section 3.3 below. Therefore, the condition $\lim_{k\to\infty} k/r_k = \infty$ is both necessary and sufficient for catastrophic overfitting: $\mathcal{E}_0 \to \infty$. Furthermore, we argue that (11) is also sufficient for benign overfitting in some settings: if $\lim_{k\to\infty} k/r_k = 0$, then we have $\lim_{n\to\infty} \mathcal{E}_0 \le (1 + \epsilon)^2$ for any $\epsilon > 0$, and thus $\mathcal{E}_0 \to 1$.

**Example 4** (Power law decay from Mallinar et al. (2022))**.**

$$\lambda_i = i^{-\alpha} \quad \text{for some } \alpha > 1.$$

In this case, we can estimate

$$\frac{1}{(\alpha - 1)(k + 1)^{\alpha - 1}} = \int_{k+1}^{\infty} x^{-\alpha}\, dx \le \sum_{i > k} \lambda_i \le \int_k^{\infty} x^{-\alpha}\, dx = \frac{1}{(\alpha - 1)k^{\alpha - 1}}$$

$$\frac{1}{(2\alpha - 1)(k + 1)^{2\alpha - 1}} = \int_{k+1}^{\infty} x^{-2\alpha}\, dx \le \sum_{i > k} \lambda_i^2 \le \int_k^{\infty} x^{-2\alpha}\, dx = \frac{1}{(2\alpha - 1)k^{2\alpha - 1}}$$

and so

$$\left(\frac{k}{k+1}\right)(\alpha - 1) \le \frac{k}{r_k} \le \left(\frac{k}{k+1}\right)^{\alpha - 1}(\alpha - 1).$$

Therefore, we have $\lim_{k\to\infty} k/r_k = \alpha - 1$ and so $\mathcal{E}_0 \lesssim \alpha$, which agrees with Mallinar et al. (2022). We remark that the Laplace kernel and ReLU NTK restricted to the hypersphere have power law decay (Geifman et al., 2020).

### 3.3 CATASTROPHIC OVERFITTING

We first state a generic non-asymptotic lower bound on $\mathcal{E}_0 = \left(1 - \frac{1}{n}\sum_i \mathcal{L}_{i,0}^2\right)^{-1}$ and then discuss the implication for catastrophic overfitting as $n$ increases.

**Theorem 5.** *For any $k \geq n$, it holds that*

$$\frac{1}{n} \sum_i \mathcal{L}_{i,0}^2 \geq \frac{n}{k} \left( \frac{k-n}{k-n+r_k} \right)^2. \tag{12}$$

For any $\epsilon > 0$, if $r_k = o(k)$ and we consider $k = (1+\epsilon)n$, then it is straightforward from (12) that $\lim_{n \to \infty} \frac{1}{n} \sum_i \mathcal{L}_{i,0}^2 \geq (1+\epsilon)^{-1}$. Since the choice of $\epsilon$ is arbitrary, we have $\lim_{n \to \infty} \frac{1}{n} \sum_i \mathcal{L}_{i,0}^2 = 1$ and so $\mathcal{E}_0 \to \infty$.

**Example 5** (Exponential decay)**.**

$$\lambda_i = e^{-i}.$$

In this case, we can estimate

$$\sum_{i>k} \lambda_i \leq \int_k^\infty e^{-x} \, dx = e^{-k}$$

and $r_k \leq e$ and $r_k/k \to 0$. Theorem 5 implies that overfitting is catastrophic, as expected from Mallinar et al. (2022).

Since Theorem 3, 4 and 5 are agnostic and non-asymptotic, we can use them to obtain an elegant characterization of whether overfitting is benign, tempered, or catastrophic, resolving an open problem[3] raised by Mallinar et al. (2022):

**Theorem 6.** *Suppose that the spectrum $\{\lambda_i\}$ is fixed as $n$ increases and contains infinitely many non-zero eigenvalues.*

*(a) If $\lim_{k \to \infty} k/r_k = 0$, then overfitting is benign: $\lim_{n \to \infty} \mathcal{E}_0 = 1$.*

*(b) If $\lim_{k \to \infty} k/r_k \in (0, \infty)$, then overfitting is tempered: $\lim_{n \to \infty} \mathcal{E}_0 \in (1, \infty)$.*

*(c) If $\lim_{k \to \infty} k/r_k = \infty$, then overfitting is catastrophic: $\lim_{n \to \infty} \mathcal{E}_0 = \infty$.*

## 4 APPLICATION: INNER-PRODUCT KERNELS IN THE POLYNOMIAL REGIME

In this section, we consider KRR with inner-product kernels in the polynomial regime (Ghorbani et al., 2021; Mei et al., 2022; Misiakiewicz, 2022). Let's take the distribution of $x$ to be uniformly distributed over the hypersphere in $\mathbb{R}^d$ or the boolean hypercube. Denote $\mathcal{V}_{\leq l-1}$ to be the subspace of all polynomials of degree $\leq l-1$ and $B(d, l) = \Theta_d(d^l)$ to be the dimension of the subspace $\mathcal{V}_l$ of degree-$l$ polynomials orthogonal to $\mathcal{V}_{\leq l-1}$. Moreover, denote $P_{\leq \lfloor l \rfloor}$ to be the projection onto $\mathcal{V}_{\leq \lfloor l \rfloor}$ and $P_{> \lfloor l \rfloor}$ to be the projection onto its complement. Let $\{Y_{ks}\}_{k \geq 0, s \in [B(d,k)]}$ be the polynomial basis with respect to $\mathcal{D}$ (e.g. spherical harmonics or parity functions).

**Inner-product kernels.** Consider kernels of the form $K(x, x') = h_d(\langle x, x' \rangle / d)$, then it admits the eigendecompositon in the polynomial basis:

$$K(x, x') = \sum_{k=0}^\infty \sum_{s \in [B(d,k)]} \frac{\mu_{d,k}(h)}{B(d,k)} Y_{ks}(x) Y_{ks}(x').$$

We also expand the target in the kernel eigenbasis and define $f^*(x) := \sum_{k=0}^\infty \sum_{s \in [B(d,k)]} v_{ks} Y_{ks}(x)$. Interestingly, the eigenvalues of $K$ with respect to $\mathcal{D}$ have a block diagonal structure. The block diagonal structure is a consequence of the rotation-invariance of the distribution of $x$.

**Polynomial regime.** Consider the regime $n \asymp d^l$ where $l$ is not an integer. We will choose $k$ in Theorem 2 to include the first $\lfloor l \rfloor$ blocks. Then

$$k = \sum_{k=0}^{\lfloor l \rfloor} B(d, k) = \Theta \left( \sum_{k=0}^{\lfloor l \rfloor} d^k \right) = \Theta \left( d^{\lfloor l \rfloor} \right) = o(n).$$

---

[3]See footnote 11 in their paper. The settings they consider (e.g., clause (a) of Theorem 3.1 with $\delta > 0$) always satisfy $\tilde{R}(\hat{f}_{\delta *}) = \sigma^2$ and so $\lim_{n \to \infty} \tilde{R}(\hat{f}_0) = \lim_{n \to \infty} \mathcal{E}_0 \cdot \sigma^2$.

and

$$R_k = \frac{\left(\sum_{k>\lfloor l \rfloor} \sum_{s \in [B(d,k)]} \frac{\mu_{d,k}(h)}{B(d,k)}\right)^2}{\sum_{k>\lfloor l \rfloor} \sum_{s \in [B(d,k)]} \left(\frac{\mu_{d,k}(h)}{B(d,k)}\right)^2} \geq \frac{\left(\sum_{k>\lfloor l \rfloor} \mu_{d,k}(h)\right)^2}{\sum_{k>\lfloor l \rfloor} \mu_{d,k}(h)^2} \cdot B(d,\lceil l \rceil)$$
$$\geq B(d,\lceil l \rceil) = \Omega(d^{\lceil l \rceil}) = \omega(n).$$

Hence, the cost of overfitting is small when $l$ is bounded away from the integers. To obtain a bound on the error of the ridgeless solution, it suffices to analyze the error of the optimally regularized model, which can be easily done with uniform convergence. Using the predictions from Simon et al. (2021), we can also recover a type of uniform convergence known as "optimistic rate" (Panchenko, 2002; Srebro et al., 2010; Zhou et al., 2021), which is suitable for the square loss.

**Theorem 7.** *Fix any $k \in \mathbb{N}$ and let $\epsilon = \sqrt{(k^2 + 2kn)/n^2}$. For any $\delta \geq 0$, it holds that*

$$(1-\epsilon)\sqrt{\tilde{R}(\hat{f}_\delta)} - \sqrt{\hat{R}(\hat{f}_\delta)} \leq \sqrt{\frac{(\sum_{i>k} \lambda_i)\|\hat{f}_\delta\|_{\mathcal{H}}^2}{n}}.$$

Note that the error of the predictor $P_{\leq \lfloor l \rfloor} f^*$ is approximately

$$\sigma^2 + \sum_{k>\lfloor l \rfloor} \sum_{s \in [B(d,k)]} v_i^2 = \sigma^2 + \|P_{>\lfloor l \rfloor} f^*\|^2. \tag{13}$$

and we can tune $\delta^*$ to match the training error of $\hat{f}_{\delta^*}$ to (13) and the Hilbert norm satisfies $\|\hat{f}_\delta\|_{\mathcal{H}} \leq \|P_{\leq \lfloor l \rfloor} f^*\|_{\mathcal{H}}$ because $\hat{f}_\delta$ is Pareto-optimal. Moreover, the expected norm of the feature is

$$\sum_{k>\lfloor l \rfloor} \sum_{s \in [B(d,k)]} \frac{\mu_{d,k}(h)}{B(d,k)} = \sum_{k>\lfloor l \rfloor} \mu_{d,k}(h),$$

and so if $\|P_{\leq \lfloor l \rfloor} f^*\|_{\mathcal{H}}^2 \cdot \left(\sum_{k>\lfloor l \rfloor} \mu_{d,k}(h)\right) = o(n)$, then $\lim_{n\to\infty} \tilde{R}(\hat{f}_{\delta^*}) \leq \sigma^2 + \|P_{>\lfloor l \rfloor} f^*\|^2$. In Ghorbani et al. (2021) and Mei et al. (2022), it is shown that the above is not just an upper bound. In fact, it holds that $\lim_{n\to\infty} R(\hat{f}_0) = \sigma^2 + \|P_{>\lfloor l \rfloor} f^*\|^2$ and our application is tight in this case.

## 5 CONCLUSION

Understanding the effect of overfitting is a fundamental problem in statistical learning theory. Contrary to the traditional intuition, prior works have shown that predictors that interpolate noisy training labels can achieve nearly optimal test error when the data distribution is well-specified. In this paper, we extend these results to the agnostic case and we use them to develop a more refined understanding of benign, tempered, and catastrophic overfitting. To the best of our knowledge, our work is the first to connect the complex closed-form risk predictions and the effective rank introduced by Bartlett et al. (2020) to establish a simple and interpretable learning guarantee for KRR. As we can see in Corollary 1 and Theorem 6, the effective ranks play a crucial role in the analysis and tightly characterize the cost of overfitting in many settings.

An interesting future direction may be asking whether our results extend to other settings, such as kernel SVM, since our theory is agnostic to the target. We hope that the theory of KRR and ridge regression with Gaussian features can lead us toward a better understanding of generalization in neural networks.

### ACKNOWLEDGMENTS

This research was done as part of the NSF-Simons Sponsored Collaboration on the Theoretical Foundations of Deep Learning.

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

## A    SUPPLEMENTAL PROOFS

In the appendix, we give proofs of all results from the main text. Our proofs are very self-contained and only use elementary results such as the Cauchy-Schwarz inequality.

### A.1    UPPER BOUNDS

The main challenge for analyzing $\mathcal{E}_0$ from equation (5) is that the effective regularization $\kappa_0$ is defined by the non-linear equation (4), which does not have a simple closed-form solution. However, the following lemma can provide an estimate for $\kappa_0$ in terms of the effective rank.

**Lemma 1.** *For any $k \in \mathbb{N}$, it holds that*

$$\kappa_0 \geq \left( 1 - \frac{n}{R_k} \right) \frac{\sum_{i>k} \lambda_i}{n} \quad and \quad \kappa_0 \geq \lambda_{k+1} \left( \frac{k + r_k}{n} - 1 \right). \tag{14}$$

*Moreover, for any $k < n$, it holds that*

$$\kappa_0 < \left( 1 - \frac{k}{n} \right)^{-1} \frac{\sum_{i>k} \lambda_i}{n}.$$

*Proof.* From the Cauchy-Schwarz inequality, we show that

$$\left( \sum_{i>k} \lambda_i \right)^2 = \left( \sum_{i>k} \sqrt{\frac{\lambda_i}{\lambda_i + \kappa_0}} \sqrt{\lambda_i(\lambda_i + \kappa_0)} \right)^2$$

$$\leq \left( \sum_{i>k} \frac{\lambda_i}{\lambda_i + \kappa_0} \right) \left( \sum_{i>k} \lambda_i(\lambda_i + \kappa_0) \right)$$

$$\leq \left( \sum_{i} \frac{\lambda_i}{\lambda_i + \kappa_0} \right) \left( \sum_{i>k} \lambda_i(\lambda_i + \kappa_0) \right)$$

$$= n \left( \sum_{i>k} \lambda_i^2 + \kappa_0 \sum_{i>k} \lambda_i \right).$$

Rearranging in terms of $\kappa_0$ proves the first inequality. Moreover, it holds that

$$n = \sum_{i \leq k} \frac{\lambda_i}{\lambda_i + \kappa_0} + \sum_{i>k} \frac{\lambda_i}{\lambda_i + \kappa_0}$$

$$\geq \frac{k\lambda_{k+1}}{\lambda_{k+1} + \kappa_0} + \frac{\sum_{i>k} \lambda_i}{\lambda_{k+1} + \kappa_0}.$$

which can be rearranged to the second lower bound. Finally, observe that

$$n = \sum_{i} \frac{\lambda_i}{\lambda_i + \kappa_0} < k + \frac{\sum_{i>k} \lambda_i}{\kappa_0}$$

and rearranging concludes the proof of the last inequality.    □

In particular, when there exists $k$ such that $k = o(n)$ and $R_k = \omega(n)$, then $\kappa_0 \approx \sum_{i>k} \lambda_i/n$. Using lemma 1, we can show Theorem 2.

**Theorem 2.** *For any $k < n$, it holds that*

$$\mathcal{E}_0 \leq \left( 1 - \frac{k}{n} \right)^{-2} \left( 1 - \frac{n}{R_k} \right)_+^{-1}. \tag{8}$$

*Proof.* For any $\delta \geq 0$, by the definition (4), we have

$$n - \frac{\delta}{\kappa_\delta} = \sum_i \frac{\lambda_i}{\lambda_i + \kappa_\delta}$$

$$\leq \sum_{i \leq k} \frac{\lambda_i}{\lambda_i + \kappa_\delta} + \sum_{i > k} \frac{\sqrt{\lambda_i}}{\lambda_i + \kappa_\delta} \sqrt{\lambda_i}$$

$$\leq k + \sqrt{\sum_{i > k} \frac{\lambda_i}{(\lambda_i + \kappa_\delta)^2} \sum_{i > k} \lambda_i}.$$

Rearranging, we get

$$\frac{\left(n - k - \frac{\delta}{\kappa_\delta}\right)^2}{\sum_{i > k} \lambda_i} \leq \sum_{i > k} \frac{\lambda_i}{(\lambda_i + \kappa_\delta)^2}. \tag{15}$$

At the same time, we can use the definition (4) again and (15) to show that

$$1 - \frac{1}{n} \sum_i \mathcal{L}_{i,\delta}^2 = \frac{1}{n} \sum_i \left[ \frac{\lambda_i}{\lambda_i + \kappa_\delta} - \left( \frac{\lambda_i}{\lambda_i + \kappa_\delta} \right)^2 \right] + \frac{\delta}{n\kappa_\delta}$$

$$= \frac{\kappa_\delta}{n} \sum_i \frac{\lambda_i}{(\lambda_i + \kappa_\delta)^2} + \frac{\delta}{n\kappa_\delta} \tag{16}$$

$$\geq \frac{\kappa_\delta}{n} \frac{\left(n - k - \frac{\delta}{\kappa_\delta}\right)^2}{\sum_{i > k} \lambda_i} + \frac{\delta}{n\kappa_\delta}.$$

Plugging in $\delta = 0$ and Lemma 1, we have

$$\mathcal{E}_0 = \left( 1 - \frac{1}{n} \sum_i \mathcal{L}_{i,0}^2 \right)^{-1} \leq \left( \frac{\kappa_0}{n} \frac{(n - k)^2}{\sum_{i > k} \lambda_i} \right)^{-1} = \left( 1 - \frac{k}{n} \right)^{-2} \left( 1 - \frac{n}{R_k} \right)^{-1}$$

provided that $R_k > n$. $\qquad \square$

Using the second part of equation (14), we can show a similar bound that depends $r_k$, which is smaller than $R_k$, but has a better dependence on $k$.

**Theorem 8.** *For any $k < n$, it holds that*

$$\mathcal{E}_0 \leq \left( 1 - \frac{k}{n} \right)^{-1} \left( 1 - \frac{n}{k + r_k} \right)_+^{-1}.$$

*Proof.* For $i \geq k + 1$, it holds that $\lambda_i \leq \lambda_{k+1}$ and so by Lemma 1, we have

$$\frac{\kappa_0}{\lambda_i + \kappa_0} \geq \frac{\kappa_0}{\lambda_{k+1} + \kappa_0} \geq \frac{\frac{k + r_k}{n} - 1}{\frac{k + r_k}{n}} = 1 - \frac{n}{k + r_k}.$$

Finally, by equation (4), we have

$$\mathcal{E}_0^{-1} = \frac{1}{n} \sum_i \frac{\lambda_i}{\lambda_i + \kappa_0} \frac{\kappa_0}{\lambda_i + \kappa_0}$$

$$\geq \frac{1}{n} \sum_{i \geq k+1} \frac{\lambda_i}{\lambda_i + \kappa_0} \frac{\kappa_0}{\lambda_i + \kappa_0}$$

$$\geq \left( 1 - \frac{k}{n} \right) \left( 1 - \frac{n}{k + r_k} \right).$$

Taking the inverse on both hand side concludes the proof. $\qquad \square$

Finally, we prove Theorem 4. The proof goes through a different argument to avoid the dependence on $1 - k/n$ because we might need to choose $k = \Omega(n)$ when overfitting is tempered.

**Theorem 4.** *Fix any $\epsilon \in (0, n/r_0)$ and consider $k_l, k_u \in \mathbb{N}$ given by*

$$k_l := \max\{k \geq 0 : k + \epsilon r_k \leq n\}$$
$$k_u := \min\{k \geq 0 : k + r_k \geq (1 + \epsilon^{-1})n\}.$$

*Then it holds that*

$$\mathcal{E}_0 \leq (1 + \epsilon)^2 \cdot \max_{k_l \leq k < k_u} \left( \frac{\lambda_{k+1}}{\lambda_{k+2}} + \frac{1}{\epsilon} \frac{k+1}{r_k - 1} \right). \tag{11}$$

*Proof.* If $\epsilon \leq n/r_0$, then it is clear that $k = 0$ satisfies $k + \epsilon r_k \leq n$. It is also clear that choosing $k \geq (1 + \epsilon^{-1})n$ satisfies $k + r_k \geq (1 + \epsilon^{-1})n$ because $r_k \geq 0$. Then both $k_l$ and $k_u$ are well-defined. To show that both are finite, we observe that $k_l \leq k_l + \epsilon r_{k_l} \leq n$ by definition and $k_u \leq (1 + \epsilon^{-1})n$ because it is defined as the minimum $k$.

Next, let $k^*$ be the smallest integer such that $\lambda_{k^*} \leq \epsilon\kappa_0$. We will show that $k^*$ is also well defined and $k^* \in [k_l + 2, k_u + 1]$. Note that for any $k < n$, we can apply Lemma 1 to show

$$\epsilon\kappa_0 < \epsilon \frac{\sum_{i>k} \lambda_i}{n-k} = \frac{\epsilon r_k}{n-k} \lambda_{k+1}.$$

Therefore, by our definition of $k_l$ and $k^*$, it holds that $\lambda_{k_l+1} > \epsilon\kappa_0 \geq \lambda_{k^*}$. Since the eigenvalues are sorted, it must hold that $k^* > k_l + 1$. On the other hand, for any $k \in \mathbb{N}$, we also apply Lemma 1 to show

$$\epsilon\kappa_0 \geq \lambda_{k+1}\epsilon \left( \frac{k + r_k}{n} - 1 \right)$$

By our definition of $k_u$ and $k^*$, it holds that $\lambda_{k_u+1} \leq \epsilon\kappa_0$ and so $k^* \leq k_u + 1$. Finally, since we have $\lambda_i \leq \lambda_{k^*} \leq \epsilon\kappa_0$ for all $i \geq k^*$ and $\lambda_{k^*-1} > \epsilon\kappa_0$, we can check that

$$\mathcal{E}_0^{-1} = 1 - \frac{1}{n} \sum_i \mathcal{L}_{i,0}^2 = \frac{\kappa_0}{n} \sum_i \frac{\lambda_i}{(\lambda_i + \kappa_0)^2}$$

$$\geq \frac{\kappa_0}{n} \sum_{i \geq k^*} \frac{\lambda_i}{(\lambda_i + \kappa_0)^2}$$

$$\geq \frac{1}{(1+\epsilon)^2} \frac{1}{n\kappa_0} \sum_{i \geq k^*} \lambda_i > \frac{\epsilon}{(1+\epsilon)^2} \frac{1}{n} \frac{\sum_{i \geq k^*-1} \lambda_i - \lambda_{k^*-1}}{\lambda_{k^*-1}}$$

$$= \frac{\epsilon}{(1+\epsilon)^2} \frac{r_{k^*-2} - 1}{n}.$$

Recall that $k^* - 1 \geq k_l + 1$ and so by definition of $k_l$, we have $k^* - 1 + \epsilon r_{k^*-1} > n$. Therefore, it holds that

$$\mathcal{E}_0 < \frac{(1+\epsilon)^2}{\epsilon} \frac{k^* - 1 + \epsilon r_{k^*-1}}{r_{k^*-2} - 1}$$

$$= (1+\epsilon)^2 \left[ \frac{\lambda_{k^*-1}}{\lambda_{k^*}} + \frac{1}{\epsilon} \frac{(k^*-2)+1}{r_{k^*-2}-1} \right].$$

where in the last step we use

$$r_{k^*-2} - 1 = \frac{\sum_{i>k^*-2} \lambda_i}{\lambda_{k^*-1}} - 1 = \frac{\sum_{i>k^*-1} \lambda_i}{\lambda_{k^*-1}}$$

$$= \frac{\lambda_{k^*}}{\lambda_{k^*-1}} r_{k^*-1}.$$

The rest follows from the fact that $k^* - 2 \in [k_l, k_u - 1]$. $\square$

## A.2 LOWER BOUNDS

We will now prove two lower bound for $\mathcal{E}_0$.

**Theorem 3.** *Fix any $b > 0$. If there exists $k < n$ such that $n \leq k + br_k$, then let $k$ be the first such integer. Otherwise, pick $k = n$. It holds that*

$$\frac{1}{n} \sum_i \mathcal{L}_{i,0}^2 \geq \max \left\{ \frac{1}{(b+1)^2} \left( 1 - \frac{k}{n} \right)^2 \frac{n}{R_k}, \left( \frac{b}{b+1} \right)^2 \frac{k}{n} \right\}. \tag{9}$$

*Proof.* First, suppose that there exists $k < n$ such that $n \leq k + br_k$ and let $k$ be the first such integer. Then we can rearrange $n \leq k + br_k$ into

$$\lambda_{k+1} \leq b \frac{\sum_{i>k} \lambda_i}{n-k},$$

and since $\lambda_i \leq \lambda_{k+1}$ for $i > k$, we apply the above and equation (14) of Lemma 1 to show that

$$\sum_i \mathcal{L}_{i,0}^2 \geq \sum_{i>k} \left( \frac{\lambda_i}{\lambda_i + \kappa_0} \right)^2$$

$$\geq \frac{\sum_{i>k} \lambda_i^2}{\left( b \frac{\sum_{i>k} \lambda_i}{n-k} + \frac{\sum_{i>k} \lambda_i}{n-k} \right)^2} = \frac{n}{(b+1)^2} \left( 1 - \frac{k}{n} \right)^2 \frac{n}{R_k}.$$

Moreover, by the definition of $k$, we must have $n > k - 1 + br_{k-1}$ which can be rearranged to

$$\lambda_k > b \frac{\sum_{i>k-1} \lambda_i}{n-k+1} \geq b\kappa_0$$

by equation (14) of Lemma 1 again. Then for any $i \leq k$, we have $\lambda_i \geq \lambda_k > b\kappa_0$ and so $\kappa_0 < \lambda_i/b$. Therefore, we have

$$\sum_i \mathcal{L}_{i,0}^2 \geq \sum_{i \leq k} \left( \frac{\lambda_i}{\lambda_i + \kappa_0} \right)^2 \geq k \left( \frac{b}{b+1} \right)^2.$$

Finally, if there is no such $k$, then the first inequality is trivial. Moreover, we have $n > n - 1 + br_{n-1}$ which rearranges to $\lambda_n \geq b \sum_{i>n-1} \lambda_i > b\kappa_0$. Therefore, by all $i \leq n$, we have $\lambda_i \geq \lambda_n > b\kappa_0$ and the rest of the proof is the same. $\square$

**Theorem 5.** *For any $k \geq n$, it holds that*

$$\frac{1}{n} \sum_i \mathcal{L}_{i,0}^2 \geq \frac{n}{k} \left( \frac{k-n}{k-n+r_k} \right)^2. \tag{12}$$

*Proof.* By the Cauchy-Schwarz inequality, we have

$$n = \sum_{i>k} \frac{\lambda_i}{\lambda_i + \kappa_0} + \sum_{i \leq k} \frac{\lambda_i}{\lambda_i + \kappa_0}$$

$$\leq \frac{\sum_{i>k} \lambda_i}{\kappa_0} + \sqrt{k} \sqrt{\sum_{i \leq k} \left( \frac{\lambda_i}{\lambda_i + \kappa_0} \right)^2}.$$

By Lemma 1, we have $\kappa_0 \geq \lambda_{k+1} \left( \frac{k+r_k}{n} - 1 \right)$. Combine with above, we obtain

$$n \leq \frac{nr_k}{k + r_k - n} + \sqrt{k} \sqrt{\sum_{i \leq k} \left( \frac{\lambda_i}{\lambda_i + \kappa_0} \right)^2}.$$

Rearranging gives us

$$\frac{n}{\sqrt{k}} \frac{k-n}{k + r_k - n} \leq \sqrt{\sum_{i \leq k} \left( \frac{\lambda_i}{\lambda_i + \kappa_0} \right)^2},$$

which implies that

$$\frac{1}{n} \sum_i \mathcal{L}_{i,0}^2 \geq \frac{1}{n} \sum_{i \leq k} \left( \frac{\lambda_i}{\lambda_i + \kappa_0} \right)^2 \geq \frac{n}{k} \left( \frac{k-n}{k + r_k - n} \right)^2.$$

$\square$

## A.3 TAXONOMY OF OVERFITTING

**Theorem 6.** *Suppose that the spectrum $\{\lambda_i\}$ is fixed as $n$ increases and contains infinitely many non-zero eigenvalues.*

*(a) If $\lim_{k \to \infty} k/r_k = 0$, then overfitting is benign: $\lim_{n \to \infty} \mathcal{E}_0 = 1$.*

*(b) If $\lim_{k \to \infty} k/r_k \in (0, \infty)$, then overfitting is tempered: $\lim_{n \to \infty} \mathcal{E}_0 \in (1, \infty)$.*

*(c) If $\lim_{k \to \infty} k/r_k = \infty$, then overfitting is catastrophic: $\lim_{n \to \infty} \mathcal{E}_0 = \infty$.*

*Proof.* We will show each clause separately.

(a) For any $\epsilon > 0$, we can pick $k = \epsilon n$ in Theorem 2 and obtain the following:

$$\mathcal{E}_0 \leq \frac{1}{(1 - \epsilon)^2} \left( 1 - \frac{1}{\epsilon} \frac{k}{R_k} \right)^{-1}.$$

Since we have

$$\sum_{i > k} \lambda_i^2 \leq \lambda_{k+1} \sum_{i > k} \lambda_i \implies R_k \geq r_k,$$

we can send $n \to \infty$ and $k/R_k \leq k/r_k \to 0$. Therefore, it holds that

$$\lim_{n \to \infty} \mathcal{E}_0 \leq \frac{1}{(1 - \epsilon)^2}.$$

Since the choice of $\epsilon > 0$ can be made arbitrarily small, we have the desired conclusion by taking $\epsilon \to 0$.

(b) If $\{k/r_k\}$ converges to a non-zero constant, then the sequence must be bounded. In particular, there exists $M > 0$ such that $r_k < kM$ for all $k$. If we let $b = 1/(3M)$ in Theorem 3, then for all $k \leq n/2$, it holds that

$$k + b r_k < k(1 + bM) \leq \frac{1 + bM}{2} n \leq \frac{2n}{3} < n.$$

Then we need to choose $k > n/2$ in Theorem 3 and

$$\frac{1}{n} \sum_i \mathcal{L}_{i,0}^2 \geq \frac{1}{2(1 + 3M)^2}$$

and so $\lim_{n \to \infty} \mathcal{E}_0 > 1$.

Similarly, there also exists $m > 0$ such that $r_k > mk$ for all $k$. Then by choosing $k = \sqrt{\frac{1}{1+m}} n$ and Theorem 8, we have

$$\mathcal{E}_0 \leq \left( 1 - \frac{k}{n} \right)^{-1} \left( 1 - \frac{1}{1+m} \frac{n}{k} \right)^{-1} = \left( 1 - \frac{1}{\sqrt{1+m}} \right)^{-2} < \infty.$$

(c) We will apply Theorem 5. For any $\epsilon > 0$, choose $k = (1 + \epsilon)n$, we get

$$\frac{1}{n} \sum_i \mathcal{L}_{i,0}^2 \geq \frac{1}{1 + \epsilon} \left( 1 - \frac{r_k}{k} \frac{1 + \epsilon}{\epsilon} \right)^2$$

Therefore, if $r_k = o(k)$, then

$$\lim_{n \to \infty} \frac{1}{n} \sum_i \mathcal{L}_{i,0}^2 \geq \frac{1}{1 + \epsilon}$$

However, since the choice of $\epsilon$ is arbitrary, then we can send $\epsilon \to 0$. The desired conclusion follows by $\mathcal{E}_0 = \left( 1 - \frac{1}{n} \sum_i \mathcal{L}_{i,0}^2 \right)^{-1}$.

$\square$

**Remark 1.** As mentioned in the main text, it is also possible to use Theorem 4 to show the upper bounds in the proof of Theorem 6 above. For simplicity, we use a different argument here by applying Theorem 2 and 8.

## B   UNIFORM CONVERGENCE

In this appendix, we show that the predictions from Simon et al. (2021) can establish a type of uniform convergence guarantee known as "optimistic rate" (Panchenko, 2002; Srebro et al., 2010) along the ridge path, which maybe of independent interest. We briefly mention the uniform convergence result in section 4 of the main text.

In particular, the tight result from Zhou et al. (2021) avoids any hidden multiplicative constant and logarithmic factor present in previous works and can be used to establish benign overfitting. However, their proof techniques depend on the Gaussian Minimax Theorem (GMT) and are limited to the setting of Gaussian features. We recover their result in Theorem 7 here with a (non-rigorous) calculation that extends beyond the Gaussian case.

### B.1   FORMULA FOR TRAINING ERROR AND HILBERT NORM

We first provide closed-form expression for the training error and Hilbert norm of $\hat{f}_\delta$. By the predictions from Simon et al. (2021), we know that

$$\hat{R}(\hat{f}_\delta) = \frac{\delta^2}{n^2 \kappa_\delta^2} \tilde{R}(\hat{f}_\delta)$$

and we can use section 4.1 of Simon et al. (2021) to compute the expected Hilbert norm:

$$
\begin{aligned}
\mathbb{E}\,\|\hat{f}_\delta\|_{\mathcal{H}}^2 &= \sum_i \frac{\mathbb{E}[\hat{v}_i^2]}{\lambda_i} = \sum_i \frac{\mathbb{E}[\hat{v}_i]^2 + \mathrm{Var}[\hat{v}_i]}{\lambda_i} \\
&= \sum_i \frac{\mathcal{L}_{i,\delta}^2 v_i^2 + \frac{\mathcal{L}_{i,\delta}^2 \tilde{R}(\hat{f}_\delta)}{n}}{\lambda_i} \\
&= \sum_i \frac{\mathcal{L}_{i,\delta}^2 v_i^2}{\lambda_i} + \frac{\tilde{R}(\hat{f}_\delta)}{n} \sum_i \frac{\mathcal{L}_{i,\delta}^2}{\lambda_i}.
\end{aligned}
$$

Therefore, we will just use the expression:

$$\|\hat{f}_\delta\|_{\mathcal{H}}^2 = \sum_i \frac{\lambda_i v_i^2}{(\lambda_i + \kappa_\delta)^2} + \frac{\tilde{R}(\hat{f}_\delta)}{n} \sum_i \frac{\lambda_i}{(\lambda_i + \kappa_\delta)^2}. \tag{17}$$

### B.2   OPTIMISTIC RATE

**Theorem 7.** *Fix any $k \in \mathbb{N}$ and let $\epsilon = \sqrt{(k^2 + 2kn)/n^2}$. For any $\delta \geq 0$, it holds that*

$$(1 - \epsilon)\sqrt{\tilde{R}(\hat{f}_\delta)} - \sqrt{\hat{R}(\hat{f}_\delta)} \leq \sqrt{\frac{(\sum_{i>k} \lambda_i)\|\hat{f}_\delta\|_{\mathcal{H}}^2}{n}}.$$

*Proof.* Applying equation (6) and (4), we can write the difference

$$
\begin{aligned}
\sqrt{\tilde{R}(\hat{f}_\delta)} - \sqrt{\hat{R}(\hat{f}_\delta)} &= \left(1 - \frac{\delta}{n\kappa_\delta}\right)\sqrt{\tilde{R}(\hat{f}_\delta)} \\
&\leq \left(\frac{1}{n}\sum_i \frac{\lambda_i}{\lambda_i + \kappa_\delta}\right)\sqrt{\tilde{R}(\hat{f}_\delta)}.
\end{aligned}
$$

By the Cauchy-Schwarz inequality, for any $k \in \mathbb{N}$, we have

$$
\left( \sum_i \frac{\lambda_i}{\lambda_i + \kappa_\delta} \right)^2 \leq \left( k + \sum_{i>k} \frac{\lambda_i}{\lambda_i + \kappa_\delta} \right)^2
$$

$$
= k^2 + 2k \left( \sum_{i>k} \frac{\lambda_i}{\lambda_i + \kappa_\delta} \right) + \left( \sum_{i>k} \frac{\sqrt{\lambda_i}}{\lambda_i + \kappa_\delta} \sqrt{\lambda_i} \right)^2
$$

$$
\leq k^2 + 2kn + \left( \sum_{i>k} \frac{\lambda_i}{(\lambda_i + \kappa_\delta)^2} \right) \left( \sum_{i>k} \lambda_i \right)
$$

By the expression (17), we have

$$
\left( \sqrt{\tilde{R}(\hat{f}_\delta)} - \sqrt{\hat{R}(\hat{f}_\delta)} \right)^2 \leq \frac{k^2 + 2kn}{n^2} \tilde{R}(\hat{f}_\delta) + \left( \frac{\tilde{R}(\hat{f}_\delta)}{n} \sum_{i>k} \frac{\lambda_i}{(\lambda_i + \kappa_\delta)^2} \right) \left( \frac{1}{n} \sum_{i>k} \lambda_i \right)
$$

$$
\leq \frac{k^2 + 2kn}{n^2} \tilde{R}(\hat{f}_\delta) + \frac{\|\hat{f}_\delta\|_{\mathcal{H}}^2 (\sum_{i>k} \lambda_i)}{n}
$$

then using $\sqrt{x+y} \leq \sqrt{x} + \sqrt{y}$, we show that

$$
\sqrt{\tilde{R}(\hat{f}_\delta)} - \sqrt{\hat{R}(\hat{f}_\delta)} \leq \sqrt{\frac{k^2 + 2kn}{n^2} \tilde{R}(\hat{f}_\delta) + \frac{\|\hat{f}_\delta\|_{\mathcal{H}}^2 (\sum_{i>k} \lambda_i)}{n}}
$$

$$
\leq \sqrt{\frac{k^2 + 2kn}{n^2} \tilde{R}(\hat{f}_\delta)} + \sqrt{\frac{\|\hat{f}_\delta\|_{\mathcal{H}}^2 (\sum_{i>k} \lambda_i)}{n}}.
$$

Rearranging concludes the proof. $\square$

### B.3 Norm Analysis

**Theorem 9.** *For any $l \in \mathbb{N} \cup \{\infty\}$ and $k \in \mathbb{N}$ such that $R_k > n$, it holds that*

$$
\|\hat{f}_0\|_{\mathcal{H}}^2 \leq \sum_{i \leq l} \frac{v_i^2}{\lambda_i} + \left( 1 - \frac{n}{R_k} \right)^{-1} \frac{n \left( \sigma^2 + \sum_{i>l} v_i^2 \right)}{\sum_{i>k} \lambda_i}.
$$

*Proof.* When $\delta = 0$, it holds that

$$
\frac{n}{\mathcal{E}_0} = n - \sum_i \mathcal{L}_{i,0}^2 = \sum_i \frac{\lambda_i}{\lambda_i + \kappa_0} - \frac{\lambda_i^2}{(\lambda_i + \kappa_0)^2}
$$

$$
= \kappa_0 \left( \sum_i \frac{\lambda_i}{(\lambda_i + \kappa_0)^2} \right)
$$

by applying (5) and (4). Therefore, the second term in (17) can be simplified as

$$
\frac{\tilde{R}(\hat{f}_0)}{n} \sum_i \frac{\lambda_i}{(\lambda_i + \kappa_0)^2} = \frac{\mathcal{E}_0 \left( \sum_i (1 - \mathcal{L}_{i,0})^2 v_i^2 + \sigma^2 \right)}{n} \frac{n}{\mathcal{E}_0 \kappa_0}
$$

$$
= \sum_i \frac{(1 - \mathcal{L}_{i,0})^2}{\kappa_0} v_i^2 + \frac{\sigma^2}{\kappa_0}
$$

$$
= \sum_i \frac{\kappa_0}{(\lambda_i + \kappa_0)^2} v_i^2 + \frac{\sigma^2}{\kappa_0}
$$

by the definition in (6). Plugging in, we arrive at

$$
\|\hat{f}_0\|_{\mathcal{H}}^2 = \sum_i \frac{v_i^2}{\lambda_i + \kappa_0} + \frac{\sigma^2}{\kappa_0}
$$

To handle situations where $f^*$ is not in the RKHS, observe that for any $l$, we have

$$\sum_i \frac{v_i^2}{\lambda_i + \kappa_0} = \sum_{i \le l} \frac{v_i^2}{\lambda_i + \kappa_0} + \sum_{i > l} \frac{v_i^2}{\lambda_i + \kappa_0}$$

$$\le \sum_{i \le l} \frac{v_i^2}{\lambda_i} + \frac{1}{\kappa_0} \sum_{i > l} v_i^2$$

and so

$$\|\hat{f}_0\|_{\mathcal{H}}^2 \le \sum_{i \le l} \frac{v_i^2}{\lambda_i} + \frac{1}{\kappa_0} \left( \sigma^2 + \sum_{i > l} v_i^2 \right).$$

The proof concludes by plugging in Lemma 1. $\qquad\square$

Finally, we can plug in the norm bound of Theorem 9 into Theorem 7 to establish benign overfitting, as in Koehler et al. (2021); Zhou et al. (2022).

**Corollary 2.** *For any $l \in \mathbb{N} \cup \{\infty\}$ and $k \in \mathbb{N}$ such that $(k/n)^2 + 2(k/n) < 1$ and $R_k > n$. Let $\epsilon = \sqrt{(k^2 + 2kn)/n^2}$, then it holds that*

$$(1 - \epsilon)^2 \tilde{R}(\hat{f}_0) \le \frac{\left( \sum_{i > k} \lambda_i \right) \left( \sum_{i \le l} \frac{v_i^2}{\lambda_i} \right)}{n} + \left( 1 - \frac{n}{R_k} \right)^{-1} \left( \sigma^2 + \sum_{i > l} v_i^2 \right). \qquad (18)$$

