# OpenReview forum: "An Agnostic View on the Cost of Overfitting in (Kernel) Ridge Regression"
_ICLR.cc/2024/Conference — ICLR 2024 poster_

### Official Review · Reviewer_oHx9 · 2023-10-16

**Soundness:** 4 excellent
**Presentation:** 4 excellent
**Contribution:** 4 excellent
**Rating:** 8
**Confidence:** 5

**Summary:**

This paper studies overfitting of kernel ridge regression (KRR) through the so-called prediction of the cost-of-overfitting:
$$
\tilde{C}(\mathcal{D},n) = \frac{\tilde{R}(\hat{f}\_{0})}{\inf\_{\delta>0}\tilde{R}(\hat{f}\_{\delta})}
$$
where the input has size $n$ drawn from the distribution $\mathcal{D}$ ; and $\hat{f}\_\delta$ denotes the KR regressor with ridge $\delta>0$; $\tilde{R}(\hat{f}\_{\delta})$ is an estimation of the KRR test error. Hence the metric $\tilde{C}(\mathcal{D},n)$ measures the ratio of the test error of the RKHS interpolant to the optimal regularised regressor and hence has range $(1,\infty]$.

This paper derives non-asymptotic bounds on the cost-of-overfitting which are independent to target function and hence agnostic. As $n\to\infty$, these bounds recover the classification result of overfitting from [Mallinar2022].

The paper is built upon the (maybe too idealised) Gaussian Design ansatz and the test error estimation result from [Simon2021], and aims to shade light to realistic scenarios of kernel training.

**Strengths:**

Originality: This paper is the first paper combining the insight from [Simon2021] and the idea from [Barlett2020] to derive a refined proof for [Mallinar2022].

Quality: The paper offers an elegant proof with detailed motivation for the definitions and intuition for the formulae, and extends the results from previous work. I have checked the proofs in details. It seems that they are all correct and I cannot see any typos from the paper.

Clarity: Definitions, theorems and proofs are clearly stated. Notations are standard and clean. Paper is easy to follow.

Significance: The simple and elegant argument surely serves its purpose in machine learning theory. The agnostic point of view about overfitting is also insightful for any future study.

**Weaknesses:**

The only potential flaw that I can see is the use of the result from [Simon2021]. Although the Gaussian Design Ansatz is widely accepted in machine learning theory, the equation (10) comes from [Simon2021] which used the rotational invariance of Gaussian, which seems to be too strong for realistic case. I wonder, if the argument of this paper can still be applicable in more realistic setup.

As stated in the abstract :"We analyze the cost of overfitting under a Gaussian universality ansatz using recently derived (non-rigorous) risk estimates in terms of the task eigenstructure." I guess that this is addressing the use of the result from [Simon2021], which was rejected by a conference before.

**Questions:**

I think the weakness I stated above is unavoidable. But if we do not focus on the cost-of-overfitting but its prediction (definition 1), every proof is rigorous and correct. The only question would be the non-rigorous prediction $C(\mathcal{D},n)\approx\tilde{C}(\mathcal{D},n)$. It would be nice if there is at least some experimental results supporting it.

---

> ### Author Response · Authors · 2023-11-18
>
> We thank the reviewer for their favorable evaluation!
>
> Regarding the reliance on the results of Simon et al. (2021): We note that the framework we take as our starting point is not only due to them: the KRR generalization equations we use (Eqs. 8-10 in our work) are obtained also in the seven other works cited at the top of Section 2.3, all of which have appeared in major peer-reviewed venues. Some of these, including Hastie et al. (2019), permit the data to be slightly different from Gaussian (e.g., covariates independent and *sub*-Gaussian), which is a bit weaker than a strict Gaussian design ansatz, but still a strong assumption. We will reword to make it clearer that these works are also sources for the framework.
>
> Regarding the use of the Gaussian design ansatz (GDA) at all (or perhaps some slightly weaker cousin): while we agree that one would ideally avoid its use, the GDA has become a standard part of the ML theory toolkit because (a) predictions generated using the GDA often match real, non-Gaussian experiment remarkably well, and (b) the GDA permits one to obtain sharp closed-form expressions in many cases. To our knowledge, no starting assumption for the feature distribution of KRR which is both more realistic and equally tractable is known.
>
> To the reviewer’s question regarding experimental results for the Gaussian equivalence approximation, we note that prior works give thorough empirical verification of the claim that $R(\hat{f_\delta}) \approx \tilde{R}(\hat{f}_\delta)$. Among those we cite at the top of Section 2.3, in particular, Jacot et al (2020), Canatar et al (2021), Simon et al. (2021), and Wei et al (2021) give experimental results for this claim.

---

### Official Review · Reviewer_avrM · 2023-11-10

**Soundness:** 3 good
**Presentation:** 3 good
**Contribution:** 3 good
**Rating:** 6
**Confidence:** 2

**Summary:**

This paper focuses on focuses on studying the cost of overfitting in noisy kernel ridge regression. The authors define this cost as the ratio of the test error of the interpolating ridgeless model to the test error of the optimally-tuned model. The approach is agnostic and provides a more refined characterization of benign, tempered, and catastrophic overfitting.

**Strengths:**

The authors offer an intricate and comprehensive examination of closed-form risk estimates in kernel ridge regression (KRR), along with a nuanced analysis of the conditions that lead to overfitting being benign, tempered, or catastrophic. The utilization of Mercer’s decomposition and the application of bi-criterion optimization within the KRR framework are particularly notable aspects of the study. Additionally, the paper is well-organized, presenting its complex ideas in a coherent structure. The inclusion of specific, practical examples to illuminate the theoretical concepts significantly enhances the paper’s clarity.

**Weaknesses:**

The paper maintains a highly specialized focus, concentrating predominantly on kernel ridge regression. This narrow scope raises questions about the generalizability of its findings to other model types, such as kernel SVM, which may not align precisely with the conditions and scenarios discussed. Despite its comprehensive and in-depth theoretical analysis, a notable limitation is the absence of empirical validation. The inclusion of studies utilizing synthetic or real-world data to substantiate the theoretical claims would greatly enhance the robustness and applicability of the paper's conclusions.

**Questions:**

The insights derived about overfitting in kernel ridge regression present significant theoretical understanding. How can these insights be effectively applied or adapted to other prevalent machine learning models and algorithms, such as neural networks or support vector machines?

Your paper shares some commonalities in terms of quantitative aspects with the study by Bartlett et al. (2020). Could you elucidate the key differences in your analysis compared to Bartlett et al.'s work? Additionally, during your analytical process, what were the most significant challenges encountered, and how did these differ from the challenges faced in the Bartlett et al. (2020) study?

The paper seems to be closely related to [1], which establishes a connection between the benign overfitting observed in ridge regression and that in SGD. Could the authors comment on the relationship/difference with [1]?

[1] Zou, et al. "Benign overfitting of constant-stepsize sgd for linear regression." Conference on Learning Theory. PMLR, 2021.

---

> ### Author Response · Authors · 2023-11-18
>
> Thanks for your feedback!
>
> Regarding the focus on (kernel) ridge regression: Indeed, as with other prominent papers on interpolation learning, we focus on ridge regression.  Ridge regression is both a fundamental and important problem in its own right, and also seen by the community (and by us) as an important “base case” to study in understanding interpolation learning.  Extending the analysis from l2 regularized regression (as in this paper) to l2 regularized large-margin classification (as in SVMs) would be interesting and indeed non-trivial, in the same way that extending previous non-agnostic analysis from ridge regression to large margin classification was.
>
> Regarding the questions:
> - Our results can be applied to understand overfitting with neural networks in the kernel regime. As discussed in, e.g., Mallinar et al. (2022), prior work showed that the spectrum of ReLU NTKs has the same decay rate as the Laplace kernel, namely, power law decay. As we show in Example 4, it implies that the NTK exhibits tempered overfitting. We will add a remark on this issue.
> - Difference from Bartlett et al. (2020): The fundamental difference, which we discuss in the introduction, is that our work takes an agnostic view and directly studies the cost of overfitting by comparing an interpolating predictor to a non-overfitting ridge predictor, instead of focusing on consistency as in Bartlett. One implication of this view is that our technical requirements that allow for benign overfitting are weaker than the requirements from Bartlett et al. (see the discussion in page 5 after Theorem 2). Moreover, a major contribution of our work is the characterization of benign/tempered/catastrophic overfitting using the ratio $k/r_k$. Such a characterization did not appear in previous works. Our proof technique is different from Bartlett et al., since we rely on the estimate from Eq. (10) and they relied on a more direct analysis.
> - The work [1] characterized benign overfitting in SGD and compared it to ridge/ridgeless regression. Still, the conceptual differences between our work and Bartlett et al. which we discussed above apply also in comparison to [1].

---

### Official Review · Reviewer_mrDt · 2023-11-12

**Soundness:** 3 good
**Presentation:** 2 fair
**Contribution:** 3 good
**Rating:** 6
**Confidence:** 3

**Summary:**

This paper studies the cost of not regularizing in kernel ridge regression. Suppose you draw some data iid from a distribution, and build a kernel matrix for this data. We could then consider two different Kernel Ridge Regression (KRR) estimators that we learn from this data:
1. The KRR estimator with the best possible regularization (this minimizes the true population risk)
2. The KRR estimator with zero regularization (this minimizes training set error)

The paper asks when the second estimator, which has zero training error because it overfits the training data, has similar true population risk as the first (and statistically optimal) estimator. Specifically, the paper is interested in bounding the ratio of these risks:
$$
\text{ratio} = \frac{\text{Risk of unregularized model}}{\text{Risk of perfectly regularized model}} \geq 1
$$

The paper is interested in the _agnostic_ setting, where we do not assume that the labeled data points are generated, or even well approximated, by a KRR estimator. Specifically, suppose we fix a distribution over input features $x$ (and hence a fixed distribution over kernel matrices) and are allowed to vary the conditional distribution of labels $y | x$. Then, **what is the maximum value that the above ratio can take?**

The paper provides bounds on this maximum ratio, denoted $\mathcal{E}_0$. At the core of the paper is Theorem 6, which provides a tight characterization for which kernels have a benign, tempered, or catastrophic concern about overfitting:
- Overfitting is _benign_ if this worst-case ratio has $\mathcal{E}_0 \rightarrow 1$ as the sample size $n\rightarrow\infty$
- Overfitting is _tempered_ if this worst-case ratio has $\mathcal{E}_0 \rightarrow c>1$ as the sample size $n\rightarrow\infty$
- Overfitting is _catastrophic_ if this worst-case ratio has $\mathcal{E}_0 \rightarrow \infty$ as the sample size $n\rightarrow\infty$

Finite sample guarantees are also given throughout the paper, and those theorems are the focus of the paper. Though, Theorem 6 above, conveys that message most concisely.

The paper is purely theoretical, and has no experiments.

**Strengths:**

The paper asks and interesting questions and formalizes this question nicely. Understanding the worst-case values of this $\text{ratio}$ across all conditional distributions $y|x$ is a good framework. The question carries sufficient significance in producing a fairly thorough understanding of a fundamental ML task (KRR) with and without regularization.

The paper makes a good effort in making the story of the results pretty clear -- we start with a naturally interesting problem, formalize it mathematically, apply a reasonable approximation and assumption, and then tackle benign, tempered, and catastrophic overfitting. The fairly mathematical results are presented nicely, which short theorem statements and clearly worked out examples demonstrating the impact of the theorems. While I sometimes wish the intuitive takeaway of these theorems was more clearly discussed, the many rigorous theoretical examples were nice to see worked out.

All of this is well executed, and piles up enough for the paper to earn a weak accept.

**Weaknesses:**

### Overview
The paper has some presentation and significance issues. I'll give my opinions and thoughts at the top of this box, and back it up with evidence at the bottom of this box.

For significance, the worst-case bounds achieved in this agnostic model are tight only in unrealistic cases. Specifically, the worst-case ratio $\mathcal{E}_0$ should only be representative of the $\text{ratio}$ for a real ML task if the Bayes-Optimal estimator is the always-zero function.

It's understandable and reasonable that we give guarantees in the worst case, but the fact that the worst-case is achieved by an unrealistic setting means that I'm skeptical that this theory can precisely characterize when overfitting actually is benign or catastrophic. The result is nice either way, but any practical takeaways will further require understanding the gap between $\mathcal{E}_0$ and $\text{ratio}$. This harms significance, but is not a deal breaker.


$\phantom{.}$

The paper also suffers from a lack of clear discussion of the mathematics involved. This is felt in two key ways: there are no proof sketches or intuitive justifications for why the theorems should be true; and there are no good intuitions given for key mathematical properties of a kernel used in the theorems. The lack of proof sketches harms my confidence that the results are correct, but frankly the results feel believable, so I'm not really marking down a lot here (though the paper would be significantly improved from just a bit of discussion of what the proofs look like).

The really painful point here is about the **Effective Rank** of a matrix. On page 5, Definition 2 introduces the values $r\_k$ and $R\_k$, both called the _effective rank_ of a matrix. $r\_k$ and $R\_k$ can differ by upwards of a quadratic factor, but no intuitive understanding of the difference between these definitions is given. Further, it's not even clear why we call these "ranks" -- usually the rank of a matrix has no notion of starting to count only from the $k^{th}$ eigenvalue.

Theorems throughout the paper depend on both $r\_k$ and $R\_k$, though $r\_k$ appears more often. I have no understanding of what these value really mean, so when I see Theorem 6 on page 8, I'm not really clear what's going on. This theorem me if our kernel matrix lies in the benign or tempered or catastrophic setting, depending on the limit $\lim\_{k\rightarrow\infty} \frac{k}{r\_k}$. Is this limit a way of characterizing the rate of decay of a spectrum? What kinds of spectra ___intuitively___ make this limit large or small?

This lack of discussion really hurts the clarity of the takeaway from the paper.

---

### Evidence
1. $\mathcal{E}_0$ _only matches_ $\text{ratio}$ _if the optimal KRR estimator is the always zero estimator_:

    This follows from the top of page 5, before equation (11). This paragraph says that $\text{ratio} = \mathcal{E}_0$ if we have $v_i = 0$ for all $i=1,...,\infty$. Here, $v_i$ is the $i^{th}$ coefficient in the expansion $f^*(x) = \sum\_{i=1}^\infty v_i \phi_i(x)$ where $f^*$ is the Bayes-optimal estimator and $\phi_i$ is a function in the Mercer-theorem-decomposition of the kernel function. Essentially, since $v_i=0$ for all $i$, we get that $f^*(x)=0$ everywhere, and hence that $\mathcal{E}_0 = \text{ratio}$ only if the Bayes-Optimal estimator is always zero.

2. _There are no proof sketches_:

    This is in all of the theorems except Theorem 1, which is proven in the body of the paper. Theorems 2 and 3 on page 5 are good examples though. It's just a theorem statement with discussion of the implication of the theorem. No justification for the correctness of the theorem is given.

3. _$r\_k$ and $R\_k$ can differ by upwards of a quadratic factor._

    See page 5, just below equation (13)

4.  _Usually the rank of a matrix has no notion of starting to count only from the $k^{th}$ eigenvalue._

    This even holds for "smooth" notions of the rank of a matrix, like the intrinsic dimension $tr(K(K+\lambda I)^{-1})$ or the stable rank $\frac{\tr(A)}{\|\|A\|\|_2}$.

**Questions:**

These are really small questions or recommended edits. You don't need to respond to these; make these edits if you want to.

1. [page 2] It feels weird to call (4) representer theorem. It's really just like kernel ridge regression. In my mind, the second line of equation (7) feels more like a representer theorem. This could just be a difference of perspective though.
1. [page 3] I have no idea what spherical harmonics or the Fourier-Walsh (parity) basis are, nor how they apply here. I'd cite something here.
1. [page 3] The omiscient risk estimate and equations (8) and (9) remind me of various ideas from various papers not discussed here. You don't need to discuss them, but I figured I should mention them since they seem relevant:
    - The effective regularization constant seems similar to the parameter $\gamma$ in equation (2) on page 3 of _[Precise expressions for random projections: Low-rank approximation and randomized Newton](https://arxiv.org/pdf/2006.10653.pdf)_
    - The $\mathcal{L}\_{i,\delta}$ values seems to be exactly _ridge leverage scores_, so that for instance $\sum\_i \mathcal{L}\_{i,\delta}$ is the intrinsic dimension of the kernel. See eg _[Fast Randomized Kernel Ridge Regression with Statistical Guarantees](https://proceedings.neurips.cc/paper_files/paper/2015/file/f3f27a324736617f20abbf2ffd806f6d-Paper.pdf)_ or _[Fast Randomized Kernel Ridge Regression with Statistical Guarantees](https://arxiv.org/pdf/1511.07263.pdf)_ or _[Random Fourier Features for Kernel Ridge Regression: Approximation Bounds and Statistical Guarantees](https://arxiv.org/pdf/1804.09893.pdf)_.
1. [page 4] For reader like me, who forget what pareto-optimal is, move the parenthetical "(point on the regularization path)" earlier in the paper to the first use of the word "pareto"
1. [page 5] The statement that $k = o(n)$ and $R_k = \omega(n)$ is one condition, not two. It's not well defined to say that $R_k = \omega(n)$ without a notion of how large $k$ is. Really, this statement is "$R_k = \omega(n)$ for some $k = o(n)$", which is clearly a single condition.
1. [page 6] At the end of example 1, explicitly note that $\mathcal{E}_0 \rightarrow 1$.
1. [page 7] It's kinda weak that the tempered overfitting doesn't have any matching lower bound part. We are already given the freedom to pick a really adversarial $y|x$ distribution, so it'd feel much more satisfying if the example could fully indulge a proof that $1 < \lim_{n\rightarrow\infty} \mathcal{E}_0 < \infty$. This might be hard, but it'd be a good payoff for strengthening the paper.
1. [page 8] Theorem 6 should be MUCH EARLIER in the paper. It's really clean and is nice to glance at and understand. It would be nice to first see this limiting result, and then see the finite-sample complexity statements that make it up. We could also then always compare a finite sample complexity result back to Theorem 6 from the (eg) introduction, making everything feel more soundly tied together into a comprehensive story.
1. [page 8] I'd change the language a bit in the theorem to recall that $\mathcal{E}_0$ is a worst-case across all $y|x$:
    - _then overfitting must be benign_
    - _then overfitting can only be tempered_
    - _then overfitting can be catastrophic_

    These phrases make it clear that, eg, $\mathcal{E}_0 \rightarrow 1$ means that you must always be benign while $\mathcal{E}_0 \rightarrow 3$ means that some $y|x$ can achieve this $\text{ratio}$ of 3 but not that all $y|x$ distributions must do this.
1. [page 8, just above "inner-product kernels"] What is $\mathcal{D}$?
1. [page 8, equation (20)] What is $\mu_{d,k}$?
1. [page 8, below equation (20)] What does i mean for the eigenvalues a matrix to have block-diagonal structure? The eigenvalues don't define a matrix.
1. [page 8, equation (22)] How does this bound on $R_k$ translate to a sufficiently tight bound on $r_k$? If $R_k = \omega(n)$ then we only know that $r_k = \omega(\sqrt{n})$, right?
1. [page 8, below equation (22)] Where and why do we need $\ell$ to be bounded away from being an integer?
1. [page 9, before "conclusion"] Spell out how these limits in this last paragraph imply that you're in the benign regime.

---

> ### Author Response · Authors · 2023-11-18
>
> Thanks for your feedback!  Your comments will greatly help us in improving the paper.
>
> *Criticism*: The worst-case bounds achieved in this agnostic model are tight only in unrealistic cases.
>
> *Our Response* (elaborated below): This is not the case. They are per-instance tight much more generally, qualitatively per-instance tight in broad generality, and since they are worst-case tight they also shed light on what can be said only based on the data distribution.
>
> *Elaboration*:
> It is true that in the paragraph you mention, at the top of page 5 (we assume you meant before eq 12, rather than before eq 11), we give the signal-less case ($v_i=0$) to argue tightness.  But this is not *necessary* for tightness, it is merely the easiest argument to show the worst-case tightness in (12).  We regret that this created an impression as if this is the only situation where $\tilde{C} \geq \mathcal{E_0}$.  In fact, in the paragraph following (12) we provide a lower bound we should have emphasized much more:
> $\tilde{C} \geq \mathcal{E_0} \frac{\sigma^2}{\tilde{R}(\hat{f}_{\delta^*})}$.
>
> This shows that $\tilde{C} \rightarrow \mathcal{E_0}$ (i.e. $\mathcal{E_0}$ precisely captures the cost of overfitting) whenever $\tilde{R}(\hat{f}_{\delta^*}) \rightarrow \sigma^2$, i.e. whenever optimal tuned ridge is consistent (approaches the Bayes error).  Thus, the upper bound is tight much more generally (recall that much of the prior literature focuses on such a consistent setting, and that this is also generically the case when $n\rightarrow\infty$ while the source distribution is fixed).  We should have said this explicitly and emphasized this.
>
> It is true that when $\tilde{R}(\hat{f}_{\delta^*}) > \sigma^2$, i.e. optimal tuned ridge is far from Bayes optimality, we could have $\mathcal{E_0}$ be a loose upper bound on $\tilde{C}$.  But even in this case, $\mathcal{E_0}$ still captures the *qualitative* (if not quantitative) noisy overfitting behavior in the following way:
> * If $\mathcal{E_0} = 1$ we have benign overfitting ($\tilde{C}=1$) regardless of the target, since $\tilde{C} \leq \mathcal{E_0}$ for any target.
> * If $\mathcal{E_0} \to \infty$ and $\sigma^2>0$ then we have catastrophic overfitting ($\tilde{C}\to\infty$) regardless of the target, since $R(\hat{f}_{\delta^*}) < \infty$ and
>
> $\tilde{C} \geq \mathcal{E_0} \frac{\sigma^2}{\tilde{R}(\hat{f}_{\delta^*})}$.
>
> * If $1 < \mathcal{E_0} < \infty$ and $\sigma^2>0$, then overfitting is either benign or tempered ($1\le\tilde{C}<\infty$), since $\tilde{C} \leq \mathcal{E_0}<\infty$.
>
> All the above is independent of the target, as long as $\sigma^2>0$ (ie we are in a noisy setting, which is where we typically talk about overfitting).  Thus, $\mathcal{E_0}$ *does* capture the *qualitative* overfitting behavior (up to the indistinguishability of benign and tempered fitting in the third case).  This discussion should of course be much more explicit in the paper.
>
> Finally, according to eq (12), $\mathcal{E_0}$ is the tightest possible “agnostic” bound on $\tilde{C}$. i.e. that does not depend on the target $f_*$ but only on the data distribution. Part of the goal of the paper is to study just how much such an agnostic view can characterize overfitting behavior, and this is the answer—it’s tight in some senses (discussed above), but is inherently not tight in a generic quantitative sense (and this is just reality, it is not due to loose analysis).
>
> Disclaimer: all the above discussion is of course based on the eigenframework and valid under it.
>
> *Lack of clear discussion of the mathematics involved.*
> The formal proofs are indeed a bit technical, and follow by analyzing the estimate from Eq. (10). We will try to add more intuitive explanations in the main text.
>
> *Regarding the comment on the effective ranks.*
> In our work, we used the definitions from the seminal work of Bartlett et al. (2020), which were also used in many other works on benign overfitting. The term “effective ranks” is used in the literature to describe these quantities. We understand your criticism regarding this term, but we preferred to be consistent with prior works.  Perhaps a better way to write the limit $\lim_{k \to \infty} \frac{k}{r_k}$ that avoids $r_k$ altogether is:
> $$\lim_{k \to \infty} \frac{k \lambda_k}{\sum_{i>k} \lambda_i}$$
> This perhaps gives a better intuitive sense of how spectral decay controls $\mathcal{E_0}$ and thus overfitting behavior.
>
> We thank you for the additional questions and recommended edits, and we will address them in the final version.

---

### Official Review · Reviewer_pfdC · 2023-11-13

**Soundness:** 3 good
**Presentation:** 3 good
**Contribution:** 3 good
**Rating:** 6
**Confidence:** 2

**Summary:**

This work study the cost of over fitting in kernel ridge regression. It is reflected by the ratio of the test error without regularizer and the optimally tuned model. It provides the necessary and sufficient conditions of three types of overfitting, benign overfitting, tempered overfitting, catastrophic overfitting. The analysis is under an “agnostic” view.

**Strengths:**

1. The work is well organized. It studies three types of overfitting: benign overfitting, tempered overfitting, and catastrophic overfitting separately.

2. The work proves matching upper and lower bounds. It gives a necessary and sufficient condition, dependent on the effective ranks of the covariance matrix, $\lim_{k\rightarrow \infty} k/r_k$, to determine whether the overfitting is benign, tempered, or catastrophic. This resolves an open problem in Mallinar et al. (2022)

3. The work provides several concrete examples to exhibit how their results work.

**Weaknesses:**

1. For people unfamiliar with the literature, some concepts are hard to understand, for example,  “omniscient risk estimate” and the “cost of overfitting.” Can you provide more intuition about these parts?

2. This work focuses on the specific problem of linear ridge regression. The analysis highly depends on the concrete structure of this problem. It might be hard to generalize to other problems of interest.

3. For a purely theoretical paper, this work provides its main results directly, without mentioning the techniques or the difficulty of the proof. Highlighting the techniques used can help the readers understand this paper's technical contribution. For example, when comparing with Bartlett et al. (2020), the paper says, "Our proof technique is completely different and simpler since we have a closed-form expression." This, however, is not helpful enough because it does not provide any equation or concrete comparison to show what the closed-form expression is and where the key difference is.

**Questions:**

1. Theorem 2 and 3 depend on $R_k$, while Theorem 4, 5, 6 depend on $r_k$. What’s the connection of these results, or are they independent?

2.  Although assumption 1 is commonly used in the literature, I’m still curious whether the analysis can work for some different assumptions. What key property of the Gaussian distribution is used in the proof?

---

> ### Author Response · Authors · 2023-11-18
>
> Thanks for your feedback!
>
> Regarding the weaknesses:
> 1. Regarding giving more intuition on terms used: As we mention in footnote 2, “omniscient risk estimate” is just a term that Wei et al. (2022) gave to the estimate which we provide in Eq. (10). The “cost of overfitting” is the main new concept we introduce: it is defined formally in Definition 1, and intuitively it compares a predictor which is obtained as a result of extreme overfitting and interpolation to the “optimal” non-overfitting predictor, and thus captures the “cost” (in terms of test error blow-up) of this overfitting.
> 2. As with many other recent papers on interpolation learning (including some very high profile papers), we focus on ridge (and kernel) regression.  Ridge regression is a fundamental and important problem in its own right, and also seen by the community (and by us) as an important “base case” to study in understanding interpolation learning, even if specific techniques will not generalize to other problems.  Indeed, the specific way we bound the “cost of overfitting” might not generalize, but a major contribution here is introducing an agnostic view of studying overfitting/interpolation and defining the “cost of overfitting”, and we believe strongly that this approach is very relevant much more generally.
> 3. The proofs follow by analyzing the estimate from Eq. (10), which gives a closed-form expression for the risk. We will add a short discussion about it.
>
> Regarding the questions:
> 1. These results are not directly comparable, except for the fact that we can use the relationship $r_k \leq R_k \leq r_k^2$ which we mention in page 5 after Eq. (13).
> 2. Strict Gaussianity isn’t actually required – the same eigenframework holds for a variety of distributions that are “close enough to Gaussian” (e.g. sub-Gaussian covariates, or “random-looking” feature vectors that satisfy some delocalization property). The key intuition is that these distributions form an equivalence class in that all are asymptotically described by the same eigenframework. The Gaussian distribution just happens to be the member of this class which is easiest to work with, so it is canonically chosen as the defining member. Empirically, it appears that feature distributions in real kernel tasks also lie in this Gaussian equivalence class (or at least close to it). This seems reasonable, but it is not yet understood why this is.

---

### Meta-Review · Area_Chair_XRPA · 2023-12-06

**Metareview:**

This paper examines the impact of overfitting in noisy kernel ridge regression (KRR), quantifying it as the ratio between the test error of the interpolating ridgeless model and that of the optimally-tuned model. The analysis in this paper offers a holistic understanding of benign, tempered, and catastrophic overfitting. This paper receives unanimous support from the reviewers, and therefore I recommend acceptance.

**Justification For Why Not Higher Score:**

This paper is purely theoretical, and its audience might be limited.

**Justification For Why Not Lower Score:**

This paper receives unanimous support from the reviewers.

---

### Decision · Program_Chairs · 2024-01-16

Accept (poster)